# TRPC4 Channel Knockdown in the Hippocampal CA1 Region Impairs Modulation of Beta Oscillations in Novel Context

**DOI:** 10.3390/biology12040629

**Published:** 2023-04-21

**Authors:** Babak Saber Marouf, Antonio Reboreda, Frederik Theissen, Rahul Kaushik, Magdalena Sauvage, Alexander Dityatev, Motoharu Yoshida

**Affiliations:** 1Institute of Physiology, Medical Faculty, Otto-Von-Guericke University, 39120 Magdeburg, Germany; 2Cognitive Neurophysiology, German Center for Neurodegenerative Diseases (DZNE), 39120 Magdeburg, Germany; 3FAM Department, Leibniz Institute for Neurobiology (LIN), 39118 Magdeburg, Germany; 4Molecular Neuroplasticity Group, German Center for Neurodegenerative Diseases (DZNE), 39120 Magdeburg, Germany; 5Medical Faculty, Otto-von-Guericke University (OvGU), 39120 Magdeburg, Germany; 6Center for Behavioral Brain Sciences (CBBS), 39106 Magdeburg, Germany

**Keywords:** TRPC4, beta oscillations, novel environment task, CA1, hippocampus, memory, encoding

## Abstract

**Simple Summary:**

Memory is a fundamental cognitive function we need for everyday life. Since we process so much information each day, it is believed that our memory system selects only relevant information to encode in order to avoid overload. In this work, we study the molecular mechanism supporting this selective memory encoding. It is believed that memory encoding occurs mainly when we encounter a new unexpected condition: a novel context. In a novel context, the hippocampus, which is crucial for memory function, intensifies a specific rhythmic activity called beta oscillations. While these beta oscillations are believed to support memory encoding, the molecular mechanisms underlying this increase in beta oscillations are not yet understood. We demonstrate that a specific type of membrane ionic channel called transient receptor potential canonical 4 (TRPC4) supports the modulation of beta oscillation in the hippocampus, indicating that TRPC4 channels play a role in novelty-related memory encoding.

**Abstract:**

Hippocampal local field potentials (LFP) are highly related to behavior and memory functions. It has been shown that beta band LFP oscillations are correlated with contextual novelty and mnemonic performance. Evidence suggests that changes in neuromodulators, such as acetylcholine and dopamine, during exploration in a novel environment underlie changes in LFP. However, potential downstream mechanisms through which neuromodulators may alter the beta band oscillation in vivo remain to be fully understood. In this paper, we study the role of the membrane cationic channel TRPC4, which is modulated by various neuromodulators through G-protein-coupled receptors, by combining shRNA-mediated TRPC4 knockdown (KD) with LFP measurements in the CA1 region of the hippocampus in behaving mice. We demonstrate that the increased beta oscillation power seen in the control group mice in a novel environment is absent in the TRPC4 KD group. A similar loss of modulation was also seen in the low-gamma band oscillations in the TRPC4 KD group. These results demonstrate that TRPC4 channels are involved in the novelty-induced modulation of beta and low-gamma oscillations in the CA1 region.

## 1. Introduction

The hippocampus plays an essential role in memory [1,2] such as episodic memory and spatial memory [3,4,5]. The hippocampal cornu ammonis 1 (CA1) region is known to be crucial for novelty detection [6], which is an essential process that allows the hippocampus to selectively encode novel information that has not been previously encountered [7]. Various studies demonstrated the pivotal role of the hippocampus in discovering and encoding novel environments and novel objects [1,2]. However, molecular mechanisms supporting novelty detection remain largely unknown.

Hippocampal neuronal oscillations in local field potentials (LFP) are known to correlate with memory functions [8,9]. The beta (~20–30 Hz) and low-gamma (~30–55 Hz) oscillations in the CA1 region have been suggested to play a pivotal role in spatial novelty and memory encoding of object location [10,11,12,13]. These oscillations are intensified during spatial novelty for a few minutes compared to familiar environments [11,14], and the beta oscillation power during novelty exposition is correlated with memory performance [11]. Furthermore, it has been reported that the spiking of CA1 neurons is phase-locked to both beta and low-gamma LFP oscillations during the exposure to novelty [15]. These findings suggest that the modulation of specific oscillations in the beta and low-gamma bands is crucial for novelty detection in the hippocampus.

In addition to the LFP oscillations, neuromodulators are shown to play a crucial role in novelty detection. The role of cholinergic and dopaminergic modulations has been established well during novel experiences [16,17]. Hippocampal acetylcholine level is raised in novel contexts [18], and cholinergic blockade impairs the encoding of object location memory [17]. Potential cellular mechanisms for preferential encoding of novel experiences have been proposed to depend on cholinergic modulations of hippocampal cellular properties [19,20]. Interestingly, in vitro studies have shown that the cholinergic receptor agonist carbachol induces beta oscillations in CA1, CA3, and dentate gyrus (DG) [21]. Similarly, carbachol also generates oscillations in the low-gamma range in vitro [22]. However, molecular mechanisms downstream of cholinergic receptor activation which support these oscillations still remain elusive.

Transient receptor potential canonical (TRPC) channels are membrane cationic channels that are modulated by neuromodulators, such as acetylcholine and dopamine, through G-protein-coupled receptors [23]. An increasing body of evidence shows the involvement of TRPC channels in the regulation of hippocampal neuronal excitability [23,24,25,26], and memory function [20,24]. The TRPC channel family consists of seven members (TRPC1-7) [27], and TRPC1, TRPC4, and TRPC5 mainly are expressed in the hippocampus, cortex, olfactory bulb, and amygdala [27]. Within the hippocampus, CA1 exhibits higher TRPC4 expression compared to CA3 and DG [28,29]. While these make the TRPC4 channel one of the potential downstream mechanisms for novelty-dependent oscillatory changes, the role of TRPC channels in the modulation of hippocampal oscillations remains largely unknown.

In the present work, we addressed the question of whether TRPC4 channels support the modulation of beta and low-gamma oscillations in a novel environment. We first established a hippocampally limited knockdown (KD) of TRPC4 channels using short hairpin RNA (shRNA) in mice. We then studied the effect of TRPC4 KD on hippocampal oscillations in a novel environment using extracellular LFP recordings with tetrodes implanted in the CA1 area while the mice explored the novel environment. We characterized the change of LFP oscillations by analyzing the power spectrum and performing a time-frequency and speed analysis. We demonstrate that novelty-related increase in oscillations is suppressed in mice with TRPC4 KD, supporting the idea that TRPC4 channels may support modulation of beta and low-gamma oscillations induced by novelty.

## 2. Methods

### 2.1. Developing TRPC4 shRNA and Scramble Viruses

TRPC4 shRNA-expressing virus was used to knock down TRPC4 channels. Oligonucleotides for TRPC4 were designed based on the study of Puram and colleagues (2011) and ordered from SIGMA-ALDRICH. The oligonucleotides used to clone siRNA (underlined below) against mTRPC4 into AAV_U6_GFP were as follows (for mTRPC4_siRNA2: siRNA_TRPC4_Mouse/Human_2: Sense and antisense):

5′-GATCCGGTCAGACTTGAACAGGCAATTCAAGAGATTGCCTGTTCAAGTCTGA

CCTTTTTTG–3′

3′-GCCAGTCTGAACTTGTCCGTTAAGTTCTCTAACGGACAAGTTCAGACTGGAAA 

AAACTTAA-5′

We used adeno-associated virus (AAV2/DJ) as vector, and U6 as promoter for this project. After purification, the AAV titers were measured using reverse transcription quantitative real-time PCR (RT-qPCR) (TRPC4_AAV_U6_shRNA2_GFP: 4.03 × 10^12^, AAV_U6_scramble_GFP: 3.56 × 10^12^) (Figure 1A). The final preparation was aliquoted in Eppendorf tubes and was transferred to −80 °C freezer for long-term storage. After developing TRPC4 shRNA virus for this purpose, the effect of shRNA-expressing virus on TRPC4 expression was verified in primary hippocampal cell cultures using RT-qPCR [30]. The cell cultures were divided into three groups (Control, Scramble, TRPC4). The control group was not infected with any virus, while the Scramble and TRPC4 KD were infected with the scramble and TRPC4 KD shRNA virus, respectively. The cell cultures were incubated at 37° C for 14 days. After 14 days, cells were collected, RNA was isolated from cells, and RT-qPCR was performed to assess the TRPC4 expression in the three groups. For the RT-qPCR, TRPC4 (Mm00444280_m1, ThermoFisher, Waltham, MA, USA) and GAPDH (Mm99999915_g1, ThermoFisher) primers were used.

### 2.2. Animals

All mice were cared for and treated strictly following the ethical animal research standards defined by the Directive of the European Communities Parliament and Council on the protection of animals used for scientific purposes (2010/63/EU). Approval was granted by the Ethical Committee on Animal Health and Care of the State of Saxony-Anhalt (TVA Az. 42502-2-1388 DZNE, 203.m-42502-2- 1665 DZNE). For the experiments, 12 weeks old male C57BL/6J mice were ordered from the DZNE animal house and transferred to the animal room at DZNE. The animals were housed together in plastic cages with closed-circuit temperature-controlled ventilation and 12/12-h light/dark phase. Mice had ad libitum access to food pellets and water. Food deprivation of up to 90 percent of animal’s current weight started simultaneously with handling the animals, to motivate animals to explore the environments to collect food. All cages, water bottles, and environment enrichment toys (running wheel) were cleaned, washed, and autoclaved before use for the animals. We used six mice that were injected with the scramble virus as the control group, and another six mice injected with the TRPC4 KD virus as the experimental group.

### 2.3. Virus Injection and Surgery

Mice were weighed and anesthetized using a ketamine and xylazine cocktail prepared in sterile tubes (ketamine 20 mg/mL + xylazine 2.5 mg/mL, 0.1 ml/20 g body weight). Around 10 min after anesthetic injection, the head was fixed in the stereotactic frame (Stoelting, equipped with a standard arm and a high-precision arm) using ear bars and nose clamp.

A small incision was made using a scalpel, and the tissues were retracted gently. Bregma was used as the reference point and the injection sites were marked according to stereotaxic coordinates (AP: −2.1, ML ± 1.7, DV: 1.1 and AP: −2.7, ML ± 2.5, DV: 1.5) [31]. Holes were drilled through the skull at the marked points with a 0.7 mm drill tip and the virus injection was performed using a Hamilton Neuros-syringe (1701, 33 g needle) and stereotaxic syringe pump (CHEMYX, NANOJET) to control injection speed and volume. The virus was injected 100 nl/min, and 1 μL was injected into each site. The needle remained at the injection site for 3–5 min before being retracted.

Next, the skin was closed using veterinary surgery glue. Following surgery, subcutaneous painkiller was injected (Carprofen, 5 mg/kg). Additional painkiller was added to the animal’s drinking water (Metamizole, 200 mg/kg). The animal was moved to a new cage with a free running wheel, and food pellets were added to the cage. Two new soft paper tissues were placed in the cage for bedding. Mice were monitored for recovery for a few hours in the surgery room and then transferred to the animal room. They were checked every day (for the next 3 days) to assess recovery and pain management.

Microdrive implants for in vivo electrophysiology were implanted in the right hemisphere and lateral to the virus injection site (AP: −2.1, ML ± 1.8, DV: 0.6). Four anchor screws and one ground screw were affixed to the skull. The implant was fixed with acrylic dental cement. To verify target sites, post-mortem histology was performed after finishing all experiments.

### 2.4. Building Tetrodes and In Vivo Electrophysiology Recording

Tetrodes were made using 12.5 μm Tungsten wire (California Fine Wire Co., Grover Beach, CA, USA). Microdrives (Axona Ltd., St Albans, UK) containing a single bundle of 8 tetrodes (32 channels) were gold plated using nanoZ (White Matter LLC, Seattle, WA, USA) in a gold solution to reduce the impedance to 150 kΩ.

After the surgery and handling of the animals, turning the screw and the screening of the neural activity to reach the CA1 pyramidal layer were performed, and the tetrodes were lowered every day based on the screening results. For the screening, open field apparatus (30 × 30 cm) was used. This open field was only used to confirm the positioning of the tetrodes in the hippocampal CA1 area. The data were recorded for 20 min. Next, the single unit recorded data were clustered using Klustakwik [32], and the separated cells were checked to see if there are place cells and to identify pyramidal cells and interneurons as well. Appearance of place cells (based on the place field map, autocorrelation, firing frequency, and waveform characteristics) was the criterion for reaching the pyramidal layer.

Electrophysiological recording and behavioral experiments were performed simultaneously. Single unit activity and local field potentials (LFP) were recorded using an in vivo multichannel data acquisition system (Axona Ltd.). Once the behavioral experiments began, data were recorded in raw mode with a sampling rate of 48 kHz.

### 2.5. Novel Environment

Four weeks after the surgery, animals were habituated to the reward pellets and were introduced to them during habituation before starting the behavioral experiment. Animals were introduced to a linear track (LT, 70 × 10 cm) for the first time to study the effect of novel context (Figure 1B).

### 2.6. Histology

Animals were perfused after finishing the experiment and brain samples were kept in −80 °C freezer after applying freeze protection protocol and snap freezing. Sections were cut with a thickness of 35 μm using a cryostat (Leica, 3050, Wetzlar, Germany). The sections were checked under a fluorescence microscope (Keyence, Osaka, Japan) for virus expression and tetrode lesions on the implanted hemisphere.

### 2.7. Data Analysis

Recorded raw data were then processed in Matlab software 2022 (MathWorks, Natick, MA, USA). LFP data were down-sampled to 2 kHz and were analyzed using a custom-written script in Matlab and Chronux toolbox.

Before performing the statistical analysis, data were checked for distribution and homogeneity using the Shapiro–Wilk test. Statistical analysis of the electrophysiological results was carried out using Matlab and SPSS (IBM). The power spectra (PS) between the groups were compared using Mann–Whitney or paired *t*-test. Mann–Whitney test was conducted for modulation index (MI) analysis between the groups. Significance level α < 0.05 (* *p* < 0.05, ** *p* < 0.01, *** *p* < 0.001) was used for all statistical analyses. Results are depicted as means ± SEM or median for non-parametric data analyses.

## 3. Results

### 3.1. TRPC4 Knockdown

To study the roles of TRPC4 channels in the hippocampal LFP oscillations during novelty exposure, we developed an shRNA-expressing virus to mitigate the expression of TRPC4 channels (please see methods). The effect of the shRNA virus was verified in primary hippocampal cell culture using RT-qPCR. There was a significant reduction in TRPC4 expression by the TRPC4 KD virus compared to the no virus control and scramble virus conditions (ANOVA with Sidak post hoc test, *p* < 0.01 and *p* < 0.05, respectively; Figure 1C). The KD virus reduced TRPC4 expression by 89.4 % relative to the scramble virus condition. Figure 1D shows an example of virus expression and tetrode placement in the hippocampus. This indicates that the TRPC4 KD virus significantly reduced TRPC4 expression, and the recordings were made from areas where the virus infected the cells. 

### 3.2. LFP Power Spectrum Analysis

The power of LFP oscillations, including theta (6–12), beta (15–30 Hz), low-gamma (30–55 Hz), and high-gamma oscillations (65–100 Hz) (Figure 2A), is known to correlate with encoding, consolidation, and retrieval of memories [5,9,33]. LFP at the CA1 pyramidal cell layer was recorded for 20 min while the animal was in the LT (Figure 2A). The overall mean power spectrum comparison between groups showed no differences for theta, beta, low-gamma, and high-gamma frequency (independent samples *t*-test, *p* = 0.513 for theta, *p* = 0.512 for beta, *p* = 0.258 for low-gamma, *p* = 0.936 for high-gamma, *p* = 0.938 for HFO) (Figure 2B). This indicates that there was no strong alteration of LFP components by TRPC4 KD.

Since the beta and low-gamma oscillations were suggested to be modulated in the order of minutes during exposure to a novel environment and during exploratory behavior [10,28], we then visualized the change of power spectrum over 20 min of exposure. It can be seen that time-frequency analyses of an example (one animal; Figure 3A) and group average (six animals in each group; Figure 3B) both show the gradual change for beta and low-gamma power in the control group. On the other hand, this is not apparent in the TRPC4 KD group (Figure 3C,D), indicating that TRPC4 KD may have prevented novelty-induced changes in beta and low-gamma oscillations.

To statistically evaluate such changes, we next compared the power of the oscillations between the first and the last five minutes of LT exposure (Figure 4A–D). Figure 4A,B shows the power spectra obtained from these two five minutes segments, and Figure 4C,D shows the magnification of the same power spectra from theta to the low-gamma band. More detailed analysis of the power of each oscillation band revealed that the theta power was increased in the scramble group but not in the KD group when the first and the last five minutes were compared (paired *t*-test, *p* < 0.001 for scramble and *p* = 0.305 for TRPC4 KD) (Figure 4E). A similar comparison indicated that the beta and low-gamma power decreased in the scramble group during the LT exposure, while the KD group maintained the power at these bands (beta, paired *t*-test, *p* < 0.05 for scramble, *p* = 0.059 for TRPC4 KD) (low-gamma, paired *t*-test, *p* < 0.05 for scramble, *p* = 0.134 for TRPC4 KD) (Figure 4F,G). This indicates that TRPC4 KD impaired the modulation of theta, beta, and low-gamma oscillations in the hippocampal CA1 area during novelty exposure. On the other hand, high-gamma and HFO band power were not modulated in either group (high-gamma, paired *t*-test, *p* = 0.630 for scramble, *p* = 0.419 for TRPC4 KD; HFO, paired *t*-test, *p* = 0.371 for scramble, *p* = 0.873 for TRPC4 KD) (Figure 4H,I). Finally, we analyzed the running speed of the mice in the corresponding two five minutes sections (paired *t*-test, *p* < 0.05 for scramble, *p* < 0.05 for TRPC4 KD) (Figure 4J). We found that both groups show a significant decrease in the running speed, indicating that the lack of modulation in the TRPC4 KD group is not purely due to different behavior in comparison to the scramble control group. These results suggest that TRPC4 channels are necessary for the novelty-induced changes in LFP oscillations in the hippocampus.

## 4. Discussion

In this study, we used an extracellular recording of LFP in the hippocampus CA1 region to study the role of TRPC4 channels in the LFP oscillations in a novel environment using a TRPC4 KD approach. After confirming the effect of the shRNA, we evaluated the change of different LFP oscillations over time while the mice explored a novel environment (linear track) for 20 min. We found that theta, beta, and low-gamma oscillation band power was modulated significantly in the scramble control group when the powers of oscillations during the first and last five minutes of exploration were compared (Figure 4A–G). In contrast, these oscillation bands were not modulated in the TRPC4 KD group.

Previous studies reported an increase in beta oscillations power in novel contexts and our results are consistent with the previous findings [10,11,12,28]. Since the increased beta oscillations (20–33 Hz) in spatial novelty coincided with the development of the spatial selectivity of place cells, it has been suggested that beta oscillation is involved in place cell learning [10], which is believed to be crucial for context representation. In addition, it has been shown that beta oscillations not only entrain the spiking activity of hippocampal neurons but also are coherent with the activity of the mid-prefrontal cortex (mPFC) and posterior parietal cortex (PAR) suggesting that beta oscillations may serve as a communication mechanism between brain regions involved in novelty detection [10]. Therefore, increased LFP oscillation may play a role both intrahippocampally and extrahippocampally.

Various functional roles of TRPC channels have been reported in different brain areas. They are engaged in neuronal excitability, excitotoxicity, neurogenesis, neurite outgrowth, and synaptic plasticity [23,26,29]. In the cognitive domain, TRPC channels are implicated in anxiety and working memory [26,27]. While the effect of TRPC channel knockout in theta-gamma cross-frequency coupling in the hippocampus has been described during REM sleep [24], the role of TRPC channels in novelty-related LFP modulation remained unknown.

TRPC channels are cationic channels present on the plasma membrane. There are four TRPC subgroups: TRPC1, TRPC2, TRPC4/5, and TRPC3/6/7 [27,29]. In situ hybridization and immunohistochemistry experiments in mice have shown that TRPC1, TRPC4, and TRPC5 genes are expressed in different hippocampal subregions [26]. In the hippocampus, TRPC1/ TRPC4 and TRPC1/ TRPC5 heteromeric channels are located in soma, dendrites, and axons [34]. They are localized in the stratum pyramidale of the hippocampal CA1-CA3 regions and the granule layer of the dentate gyrus (DG) [26].

TRPC channels are activated via a G-protein-coupled receptor-phospholipase C-dependent pathway (Figure 5) [34,35,36]. It is well known that the acetylcholine (ACh) level is increased in the hippocampus during exploration in a novel context [35,36]. In addition, beta and gamma oscillations can be induced by the increased tone of cholinergic receptor activation in hippocampal slice preparations [18]. Since the cholinergic receptor activates TRPC channels through the Gq- and Gi-protein-coupled receptors, our data suggest that the cholinergic modulation of TRPC4 may underly novelty-dependent beta and gamma oscillation changes. In vitro studies have also shown that mGluR I activation, in addition to AMPA and NMDA receptors, is involved in the modulation of beta oscillations [21,37]. Since group I mGluR also activates TRPC channels through the same Gq and Gi pathways [23,26,27], these studies are also in line with our result that TRPC4 is involved in the modulation of beta oscillations.

While it still remains unknown how TRPC channels can modulate beta and gamma oscillations, one potential mechanism is the slow depolarization and increased excitability due to TRPC activation. Both cholinergic and group I mGluR agonists, for instance (S)-3,5-dihydroxyphenylglycine (DHPG), lead to Ca^2+^ influx and membrane depolarization in hippocampal pyramidal cells [27]. This possibility agrees with the fact that cellular activity is increased during beta oscillations both in vitro and in vivo [38], and the fact that memory encoding through synaptic plasticity, which requires Ca^2+^ influx, is enhanced in novel contexts.

Finally, we also observed that the modulation of theta oscillation power was absent in the TRPC4 KD group. While modulation of theta by a novel context has not been described intensively, it has been reported that theta peak frequency drops in response to novel environments [39]. Interestingly, this drop in the theta frequency is proposed to underly the change of grid cell firing in the medial entorhinal cortex [17], suggesting a link between theta modulation and context representation similar to the case with the beta oscillations discussed above [10,40]. In addition, the importance of theta oscillations in memory encoding, which occurs predominantly in a novel condition, has intensively been studied [41]. Together, these data suggest that novelty-dependent modulation of hippocampal oscillatory activity through TRPC4 channels may contribute to the encoding of novel context.

## 5. Conclusions

In conclusion, we demonstrated, for the first time, that the modulations of hippocampal theta, beta, and low-gamma oscillations in environmental novelty were disrupted by TRPC4 KD. Our data suggest that TRPC4 channels play an important role in coupling between the neuromodulatory changes (e.g., cholinergic system) due to novelty and the hippocampal oscillatory activity in the CA1 area.

## Figures and Tables

**Figure 1 biology-12-00629-f001:**
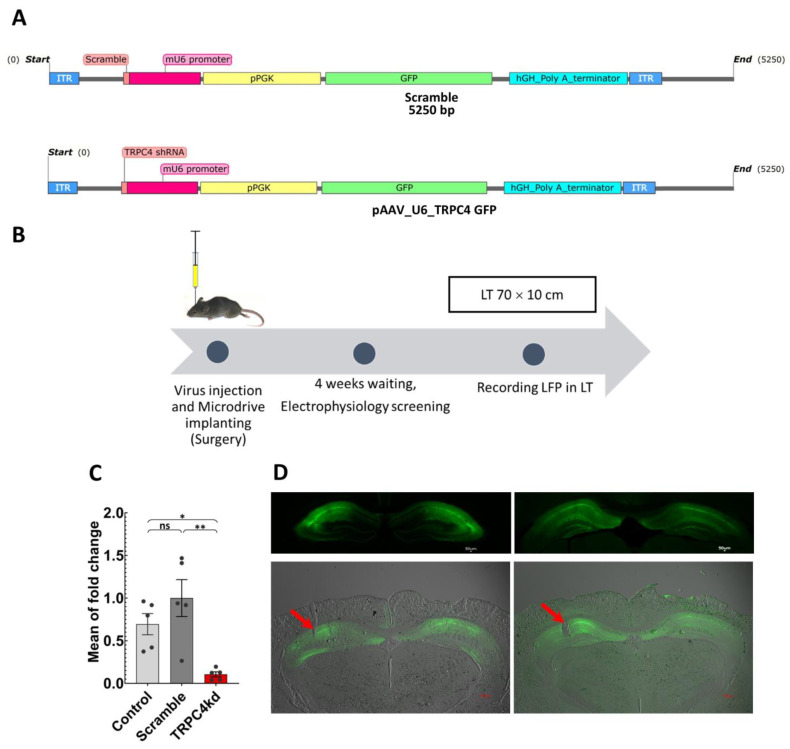
(**A**) Schematic of the AAV scramble virus (top) and TRPC4 KD virus (bottom) showing the promoter, shRNA, and GFP (SnapGene software 6.0.5 (www.snapgene.com)). (**B**) Time chart of the experimental steps from surgery until recording. (**C**) Fold change in TRPC4 expression in primary hippocampal cell culture. The TRPC4 expression was verified using RT-qPCR on day 14 after AAV delivery on day 7 (ANOVA, *p* < 0.01 between scramble and TRPC4 KD, *p* < 0.05 between control and TRPC4 KD, n = 5 for control, n = 5 for scramble, n = 5 for TRPC4 KD). (**D**) Viral expression in the mouse hippocampus. **Top**, GFP expression in the hippocampal coronal sections after scramble (Left) or TRPC4 KD AAV (Right) injection in the CA1 region. **Bottom**, Microdrive implant control. An example of the tetrodes tip lesion is visible on the left side of the images. Examples of tetrode recording sites in scramble (Left) and TRPC4 KD AAV (Right) injected mice. The red arrows show the lesions made by tetrode tips. The acronyms are LT: linear track; LFP: local field potentials. *: *p* < 0.05, **: *p* < 0.01, ns: not significant.

**Figure 2 biology-12-00629-f002:**
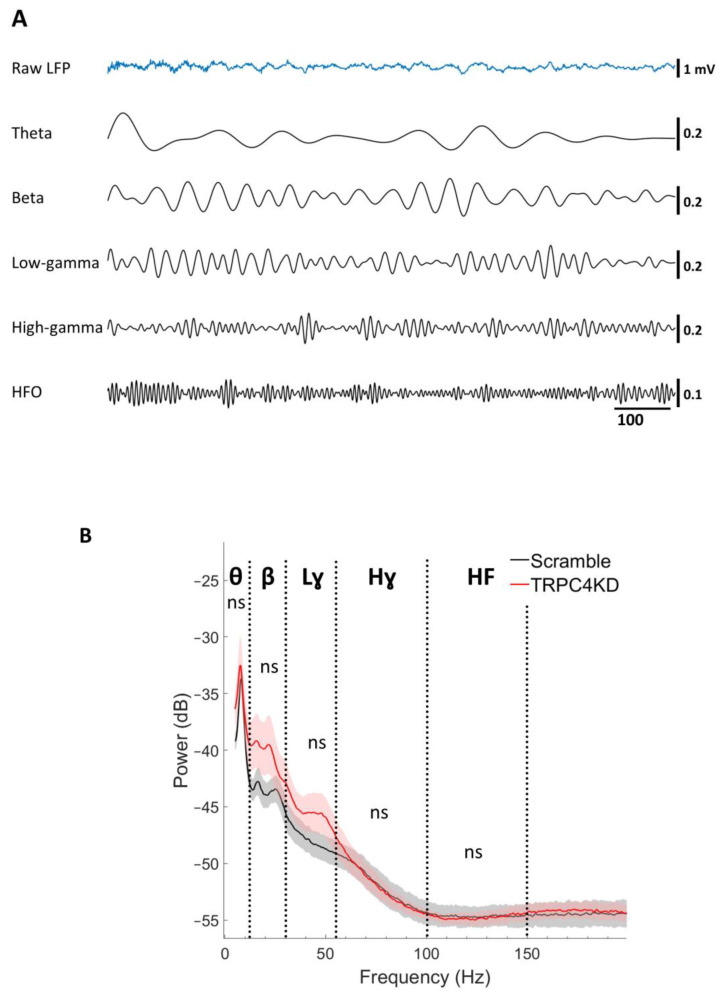
(**A**) An example of LFP and underlying theta, beta, low gamma, high gamma oscillations, and HFO in the order from top to bottom. Theta: 7-12 Hz, Beta: 15–29 Hz, Low-gamma: 30–55 Hz, High-gamma: 65–100 Hz, High-frequency oscillation (HFO): 100–150 Hz. (**B**) Mean power spectrum comparison between groups (independent samples *t*-test, *p* = 0.513 for theta, *p* = 0.512 for beta, *p* = 0.258 for low-gamma, *p* = 0.936 for high-gamma, *p* = 0.938 for HFO). ns, not significant. Scramble (black): n = 6; TRPC4 KD (red): n = 6. Filled area indicates standard error. ns: not significant.

**Figure 3 biology-12-00629-f003:**
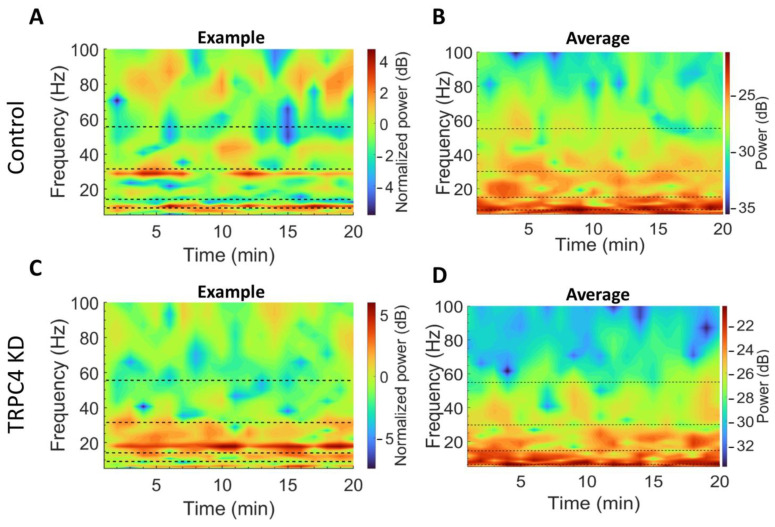
Time-frequency analysis of LFP. (**A**,**B**) Example and average spectrograms, respectively, from the scramble control group. (**C**,**D**) Example and average spectrograms, respectively, from the TRPC4 KD group. Spectrograms of individual examples (**A**,**C**) are normalized to the power at the first minute.

**Figure 4 biology-12-00629-f004:**
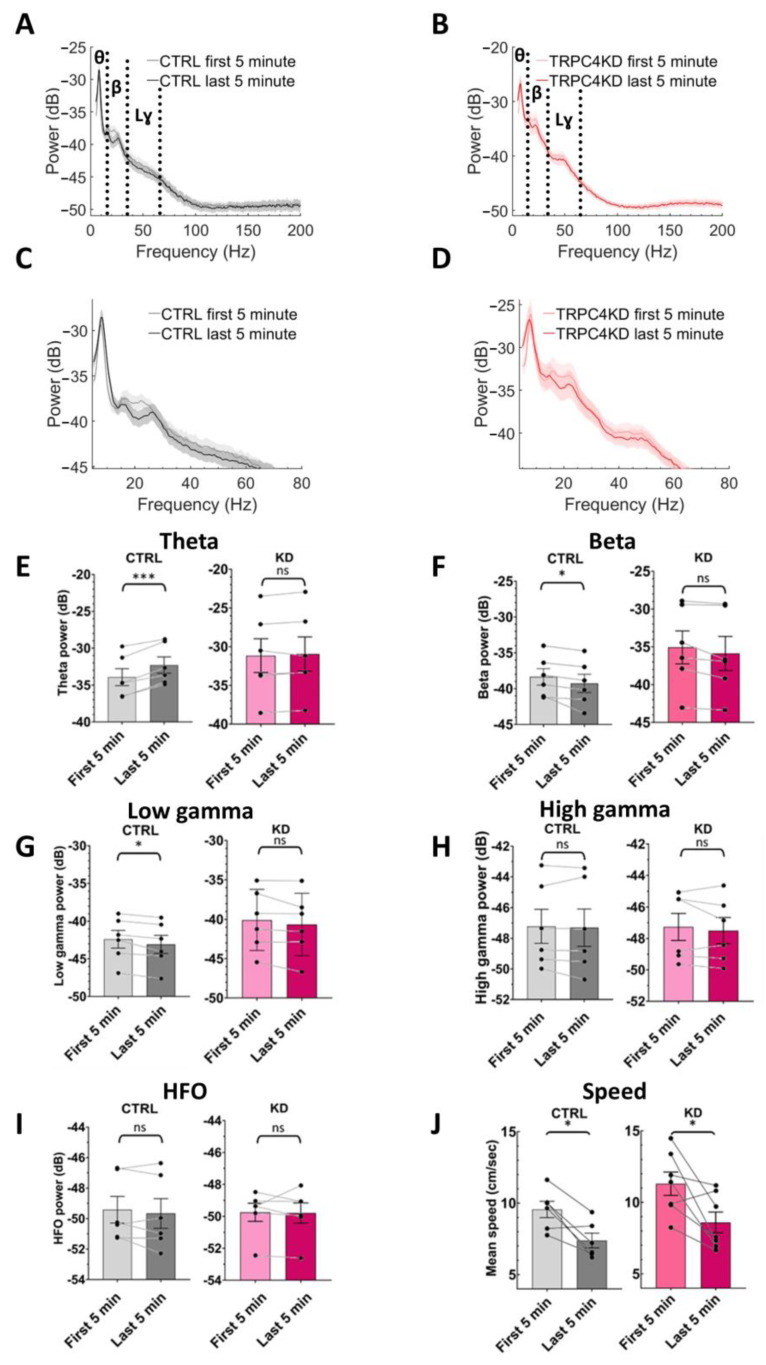
Comparison of the first and the last 5 min of the LT task. (**A**) Mean power spectra in the first 5 min and the last 5 min for scramble group. (**B**) Mean power spectra in the first 5 min and the last 5 min for TRPC4 KD group. (**C**,**D**) Rescaled power spectra shown in (**A**,**B**). (**E**) Theta oscillations (paired *t*-test, *p* < 0.001 for scramble and *p* = 0.305 for TRPC4 KD). (**F**) Beta oscillations (paired *t*-test, *p* < 0.05 for scramble, *p* = 0.059 for TRPC4 KD). (**G**) Low-gamma oscillations (paired *t*-test, *p* < 0.05 for scramble. *p* = 0.134 for TRPC4 KD). (**H**) High-gamma oscillations (paired *t*-test, *p* = 0.630 for scramble, *p* = 0.419 for TRPC4 KD). (**I**) High-frequency oscillations (paired *t*-test, *p* = 0.371 for scramble, *p* = 0.873 for TRPC4 KD); HFO: high-frequency oscillations. (**J**) Mean of speed during the first 5 and last 5 min of the task (paired *t*-test, *p* < 0.05 for scramble, *p* < 0.05 for TRPC4 KD). ns, not significant. In all panels, scramble: n = 6; TRPC4 KD, n = 6. *: *p* < 0.05, ***: *p* < 0.001, ns: not significant.

**Figure 5 biology-12-00629-f005:**
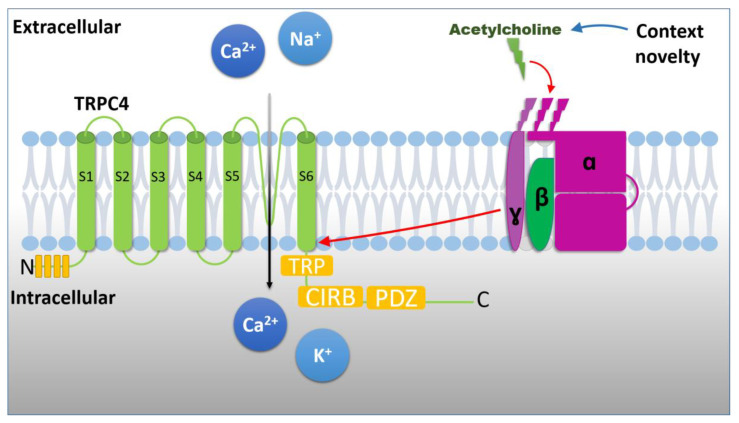
A TRPC4 channel with N terminus and C terminal has TRP, CRIB, and PDZ domains. Signal transduction through muscarinic receptors is triggered by acetylcholine neurotransmitter. Following acetylcholine binding to the receptor, TRPC4 channel is activated through G protein cascade. The acronyms are CIRB: calmodulin- and inositol triphosphate receptor (InsP3R)-binding site; PDZ: amino acid binding motif; TRP: TRP-like domain.

## Data Availability

The data that support the findings of this study are available on request from the corresponding author.

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
