# Peer review of "TRPC4 Channel Knockdown in the Hippocampal CA1 Region Impairs Modulation of Beta Oscillations in Novel Context"

_biology, 2023, doi:10.3390/biology12040629_

Round 1

Reviewer 1 Report

This is a review of the article entitled “TRPC4 channel knockdown in the hippocampal CA1 region 2 impairs modulation of beta oscillations in novel context” submitted to the MDPI biology journal. In this study, the authors have studied how a specific type of membrane ionic channels called transient receptor potential canonical 4 (TRPC4) supports the modulation of beta oscillation in the hippocampus. The hippocampus, an important memory center in the brain, intensifies a specific rhythmic activity called beta oscillation when encountering a novel and unexpected condition. While these beta oscillations are believed to support memory encoding, the molecular mechanisms underlying the increase of beta oscillation are not completely understood. This study focuses on the importance of membrane cationic channel TRCP4 in beta oscillation by combining shRNA-mediated TRPC4 knockdown (KD) with hippocampal local field potential (LFP) measurements in the CA1 regions of C57BL/6J mice. By performing a time-frequency, power spectrum, and speed analysis, authors have demonstrated that novelty-associated increase in oscillations was reduced in TRPC4 KD mice. Well-discussed and well-executed study showing that the changes of hippocampal theta, beta, and low-gamma oscillations in a novel environment were disrupted by TRPC4 KD.

Below are some comments for the authors to consider for improving the clarity of this research paper. Overall, there are minor grammatical errors that need to be fixed.

·         Introduction, Line 43. Mention the full form of CA1.

·         Introduction, Line 84. Mention the full form of shRNA.

·         Introduction, Line 86. Add the word ‘a’ before the word “tetrode” for better clarity.

·         Methods, 2.1. Spelling error, correct as “promoter”.

·         Methods, Line 107. Mention the units of temperature for the freezer.

·         Methods. Figure 1B is mentioned first and then Figure 1A is mentioned. It is better to renumber the figure subsections so that they are in sequence.

·         Methods. Mention how many control mice were used. Also, what is the rationale behind choosing only male mice for the study?

·         Methods, Line 138. Change “was retracted” to “were retracted”.

·         Methods, 2.6. Mention the size of the cryosections.

·         Methods. Include methodology for the open field test. In the open field test, did the authors measure %time in center and distance traveled to determine anxiety?

·         Results, Section 3.1, second paragraph. Move the second paragraph to the methods section, and only discuss results in this section. Also, the order of figure subsections needs to be in sequence.

·         Results. After each section, add a sentence at the end to summarize the results from that section.  

·         Figure 1A. In the figure legend, mention the full form of OF, LFP, and LT.

·         Figure 1B. The text in this figure is not clear. Increase the font size of this figure. Also, mention the full of abbreviations in the figure legend.

·         Figure 1D. Mention in the figure legend red arrow shows the tetrodes tip lesion.

·         Figure 2. Mention in the figure text what the different colored lines represent.

·         Figure 3. Label C and D with Example and Average, respectively, like A and B to improve clarity and continuity.

·         Figure 4. For Graphs A, B, C, and D, change legends, scale and axis titles to have the same font size.

·         Figure 4. Include the abbreviation HFO in the description of figure I.

·         Discussion. Are there any sex differences in the response of the hippocampus to beta oscillations in a novel environment?

Overall, there are minor grammatical errors of using the articles that need to be fixed.

Author Response

We very much thank the reviewers for the constructive comments and suggestions to enhance the quality of our manuscript. Please find below a point-by-point reply addressing the reviewers' concerns. We hope we addressed all comments adequately. Comments from the reviewers are in italics and underlined for clarity. All changes we made on the revised manuscript are tracked for clarity. 

Reviewer 1:

Comments and Suggestions for Authors

This is a review of the article entitled “TRPC4 channel knockdown in the hippocampal CA1 region 2 impairs modulation of beta oscillations in novel context” submitted to the MDPI biology journal. In this study, the authors have studied how a specific type of membrane ionic channels called transient receptor potential canonical 4 (TRPC4) supports the modulation of beta oscillation in the hippocampus. The hippocampus, an important memory center in the brain, intensifies a specific rhythmic activity called beta oscillation when encountering a novel and unexpected condition. While these beta oscillations are believed to support memory encoding, the molecular mechanisms underlying the increase of beta oscillation are not completely understood. This study focuses on the importance of membrane cationic channel TRCP4 in beta oscillation by combining shRNA-mediated TRPC4 knockdown (KD) with hippocampal local field potential (LFP) measurements in the CA1 regions of C57BL/6J mice. By performing a time-frequency, power spectrum, and speed analysis, authors have demonstrated that novelty-associated increase in oscillations was reduced in TRPC4 KD mice. Well-discussed and well-executed study showing that the changes of hippocampal theta, beta, and low-gamma oscillations in a novel environment were disrupted by TRPC4 KD.

Below are some comments for the authors to consider for improving the clarity of this research paper. Overall, there are minor grammatical errors that need to be fixed.

  •     Introduction, Line 43. Mention the full form of CA1.

Response: Changes were made accordingly.

  •     Introduction, Line 84. Mention the full form of shRNA.

Response: Changes were made accordingly.

  •     Introduction, Line 86. Add the word ‘a’ before the word “tetrode” for better clarity.

Response: We now write “tetrodes” since we used eight of them.

  •     Methods, 2.1. Spelling error, correct as “promoter”.

Response: Changes were made accordingly.

  •     Methods, Line 107. Mention the units of temperature for the freezer.

Response: Changes were made accordingly.

  •     Methods. Figure 1B is mentioned first and then Figure 1A is mentioned. It is better to renumber the figure subsections so that they are in sequence.

Response: Changes were made accordingly.

  •     Methods. Mention how many control mice were used. Also, what is the rationale behind choosing only male mice for the study?

Response: We used 6 mice in the control and 6 mice in the TRPC4 knock down group. It is mentioned in line number 132-133. We made this clearer by modifying this sentence as follows: “We used six mice that were injected with the scramble virus as the control group, and another six mice injected with the TRPC4 KD virus as the experimental group.”

We did not use female animals in this study because, to our best knowledge, the increase in beta oscillations in novel environments is not reported in female animals yet. Since our study focused on the role of TRPC4 channels in beta oscillation increase, we needed to study the role of TRPC4 channels in the male animals in which novelty induced modulation of beta oscillation is established.  

In our lab, we used female mice in other studies that are also related to TRPC channels, such as Valero-Aracama et al., Behav Brain Res. 2015.  Future studies should establish the role of beta oscillations in novelty in females, and the role of TRPC4 channels in beta oscillations should also be studied in female animals.

  •     Methods, Line 138. Change “was retracted” to “were retracted”.

Response: Changes were made accordingly.

  •     Methods, 2.6. Mention the size of the cryosections.

Response: Changes were made accordingly. We now state “Sections were cut with the thickness of 35 μm using a cryostat (Leica, 3050).”

  •     Methods. Include methodology for the open field test. In the open field test, did the authors measure %time in center and distance traveled to determine anxiety?

Response: We indeed used an open field. However, this was not a test for anxiety. It was only to test if the electrodes (tetrodes) were placed at the hippocampus using the activity measured while the animals explored in the open field. This is a routine procedure in this field. Our focus on this manuscript was on novelty, and we did not have intention to measure anxiety. Therefore, we did not measure the time spent at the center and distance traveled. We now state this in the methods section to clarify this point as follows. 

“This open field was only used to confirm the positioning of the tetrodes in the hippocampal CA1 area”.

In addition, following this comment, we removed the figure of the open field to avoid confusion.

  •     Results, Section 3.1, second paragraph. Move the second paragraph to the methods section, and only discuss results in this section. Also, the order of figure subsections needs to be in sequence.

Response: Changes were made accordingly.

  •     Results. After each section, add a sentence at the end to summarize the results from that section.  

Response: Changes were made accordingly.

  •     Figure 1A. In the figure legend, mention the full form of OF, LFP, and LT.

Response: Changes were made accordingly.

  •     Figure 1B. The text in this figure is not clear. Increase the font size of this figure. Also, mention the full of abbreviations in the figure legend.

Response: Changes were made accordingly.

  •     Figure 1D. Mention in the figure legend red arrow shows the tetrodes tip lesion.

Response: Changes were made accordingly.

  •     Figure 2. Mention in the figure text what the different colored lines represent.

Response: Changes were made accordingly.

  •     Figure 3. Label C and D with Example and Average, respectively, like A and B to improve clarity and continuity.

Response: Changes were made accordingly.

  •     Figure 4. For Graphs A, B, C, and D, change legends, scale and axis titles to have the same font size.

Response: Changes were made accordingly.

  •     Figure 4. Include the abbreviation HFO in the description of figure I.

Response: Changes were made accordingly.

  •     Discussion. Are there any sex differences in the response of the hippocampus to beta oscillations in a novel environment?

Response: To our knowledge, there is no report of beta oscillations in  female animals  in novel context. Therefore, it is difficult to discuss this point. We believe that knowledge of beta oscillation changes in novelty should be studied in females, and then whether TRPC4 channels play their role or not  should also be studied in the future.

Comments on the Quality of English Language

Overall, there are minor grammatical errors of using the articles that need to be fixed.

We read through the entire manuscript and corrected grammatical errors. 

Reviewer 2 Report

The manuscript entitled “TRPC4 channel knockdown in the hippocampal CA1 region 2 impairs modulation of beta oscillations in novel context” by Saber Marouf and colleagues is focusing a very actual and important topic. It brings some interesting and novel data. I have only a few suggestions to improve the already high quality of the manuscript.

Check the meaning of …the molecular correlate at the level of membrane ion channels that are in charge of modulating  beta band oscillation in vivo remains to be fully understood., in the abstract (lines 29-30).

In keywords, use beta oscillations as one keyword, and novelty should be rephrased as a novel environment task.

Reconsider the order of subsections in MM (data analysis at the end of the section).

Author Response

We very much thank the reviewers for the constructive comments and suggestions to enhance the quality of our manuscript. Please find below a point-by-point reply addressing the reviewers' concerns. We hope we addressed all comments adequately. Comments from the reviewers are in italics and underlined for clarity. All changes we made on the revised manuscript are tracked for clarity.

Reviewer 2:

Comments and Suggestions for Authors

The manuscript entitled “TRPC4 channel knockdown in the hippocampal CA1 region 2 impairs modulation of beta oscillations in novel context” by Saber Marouf and colleagues is focusing a very actual and important topic. It brings some interesting and novel data. I have only a few suggestions to improve the already high quality of the manuscript.

Check the meaning of …the molecular correlate at the level of membrane ion channels that are in charge of modulating  beta band oscillation in vivo remains to be fully understood., in the abstract (lines 29-30).

Response: We acknowledge the sentence might be confusing and we would like to rephrase it as follows: “However, potential downstream mechanisms through which neuromodulators may alter the beta band oscillation in vivo remain to be fully understood.”

In keywords, use beta oscillations as one keyword, and novelty should be rephrased as a novel environment task.

Response: Changes were made accordingly.

Reconsider the order of subsections in MM (data analysis at the end of the section).

Response: Changes were made accordingly.
